# Development and Optimization of Alpha-Pinene-Loaded Solid Lipid Nanoparticles (SLN) Using Experimental Factorial Design and Dispersion Analysis

**DOI:** 10.3390/molecules24152683

**Published:** 2019-07-24

**Authors:** Aleksandra Zielińska, Nuno R. Ferreira, Alessandra Durazzo, Massimo Lucarini, Nicola Cicero, Soukaina El Mamouni, Amélia M. Silva, Izabela Nowak, Antonello Santini, Eliana B. Souto

**Affiliations:** 1Department of Pharmaceutical Technology, Faculty of Pharmacy, University of Coimbra (FFUC), Pólo das Ciências da Saúde, Azinhaga de Santa Comba, 3000-548 Coimbra, Portugal; 2Faculty of Chemistry, Adam Mickiewicz University in Poznań, 61-712 Poznań, Poland; 3CREA-Research Centre for Food and Nutrition, Via Ardeatina 546, 00178 Rome, Italy; 4Dipartimento di Scienze biomediche, odontoiatriche e delle immagini morfologiche e funzionali, Università degli Studi di Messina, Polo Universitario Annunziata, 98168 Messina, Italy; 5Department of Pharmacy, University of Napoli Federico II, 80138 Napoli NA, Italy; 6Department of Biology and Environment, University of Trás-os-Montes e Alto Douro (UTAD), Quinta de Prados, P-5001-801 Vila Real, Portugal; 7Centre for Research and Technology of Agro-Environmental and Biological Sciences (CITAB), University of Trás-os-Montes e Alto Douro (UTAD), P-5001-801 Vila Real, Portugal; 8CEB - Centre of Biological Engineering, University of Minho, Campus de Gualtar, 4710-057 Braga, Portugal

**Keywords:** α-pinene, solid lipid nanoparticles, factorial design, optimization, dispersion analyzer, instability phenomena

## Abstract

The encapsulation of bicyclic monoterpene α-pinene into solid lipid nanoparticles (SLN) is reported using experimental factorial design, followed by high-end dispersion analyzer LUMiSizer^®^. This equipment allows the characterization of the α-pinene-loaded SLN instability phenomena (e.g., sedimentation, flotation or coagulation), as well as the determination of the velocity distribution in the centrifugal field and the particle size distribution. In this work, SLN were produced by hot high-pressure homogenization technique. The influence of the independent variables, surfactant and lipid ratio on the physicochemical properties of SLN, such as mean particle size (Z-Ave), polydispersity index (PDI) and zeta potential (ZP), was estimated using a 2^2^-factorial design. The Z-Ave and PDI were analyzed by dynamic light scattering, while ZP measurements were recorded by electrophoretic light scattering. Based on the obtained results, the optimal SLN dispersion was composed of 1 wt.% of α-pinene, 4 wt.% of solid lipid (Imwitor^®^ 900 K) and 2.5 wt.% of surfactant (Poloxamer 188), depicting 136.7 nm of Z-Ave, 0.170 of PDI and 0 mV of ZP. Furthermore, LUMISizer^®^ has been successfully used in the stability analysis of α-pinene-loaded SLN.

## 1. Introduction

Innovative, non-toxic lipid nanoparticles, such as solid lipid nanoparticles (SLN), are recognized as suitable delivery carriers for lipophilic active pharmaceutical ingredients (APIs) [1,2]. SLN are obtained from physiological and biodegradable lipids, classified as ‘generally recognized as safe’ (GRAS) [3,4], being mainly composed of solid lipids (e.g., triglycerides, saturated fatty acids or waxes) [5]. The prerequisite for the selection of the raw materials is their melting point (above 40 °C), because SLN have to be solid at both of the room and body temperatures [6]. The interest in SLN for loading essential oils, containing monoterpenes, relies on their capacity to modify the release profile of perfumes and fragrances [7,8], as well as their high tolerance for a topical application on the skin [1,9,10].

Monoterpenes (C_10_H_16_) are recognized to be of industrial interest, mainly in the field of pharmaceutics, nutraceuticals and cosmetics [2,11,12,13,14,15,16,17,18,19,20]. One of the most commonly well-known bicyclic monoterpenes [21] is alpha-pinene (Figure 1), also called 2,6,6-trimethylbicyclo [3.1.1] hept-2-ene [22,23]. Alpha-pinene occurs in the essential oil of several coniferous trees from *Pinaceae* (genus *Pinus*) [24] and *Lamiaceae* family (e.g., lavender, genus Lavandula) [25,26,27], rosemary (genus *Rosmarinus*, species *Rosmarinus officinalis* L.) [21,25,28]. It can also be extracted from mandarin peel oil (Rutaceae family, Citrus reticulate species) [29]. Alpha-pinene is a colorless liquid at room temperature, substantially insoluble in water, being therefore a suitable candidate for the loading in lipid nanoparticles as SLN for modified release. This monoterpene can be widely used as a raw material for the synthesis of products with a high commercial value [22,30,31] in the pharmaceutical, fragrance and flavor industries [22,23].

Alpha-pinene exhibits several biological and medical properties, e.g., antimicrobial [32,33], insecticidal or antioxidant activities [23,34]. It has anti-inflammatory [35,36], anti-stress and anti-convulsive activities, as well as sedative effects and antitumor activity [21,37,38,39,40,41]. Several products can be obtained by submitting α-pinene to different catalytic chemical processes. For instance, α-pinene oxidation produces α-pinene oxide, verbenone and verbenol, which are used in the production of artificial flavors, fragrances and medicines [42]. Other terpenes used in industry, such as β-pinene [43], tricyclene, camphene, limonene, p-cymene, terpinenes or terpinolenes are the result of α-pinene isomerization in the presence of acid catalysts [24,44]. Due to the identified beneficial effects of α-pinene, its loading in SLN may be an interesting non-toxic skin formulation.

In this study, experimental factorial design was used to develop and optimize α-pinene-loaded SLN dispersion with suitable physicochemical parameters for the encapsulation of α-pinene. Additionally, the stability of α-pinene-loaded SLN stored at room temperature was characterized by using a dispersion analyzer (LUMiSizer^®^) with STEP-Technology^®^ (Space- and Time-resolved Extinction Profiles). 

## 2. Material and Methods

### 2.1. Materials

Compritol^®^ 888 ATO, Glyceryl Dibehenate; Gattefossé (Madrid, Spain); Dynasan^®^116, Tripalmitin; Cremer Oleo GmbH & Co. KG company (Hamburg, Germany); Dynasan^®^118, Monoacid Triglyceride; Cremer Oleo GmbH&Co. KG company (Hamburg, Germany); Dynasan^®^ P 60 (F), Palmitic/Stearic Triglycerides; Cremer Oleo GmbH&Co. KG company (Hamburg, Germany); Imwitor^®^ 900 K, Glycerol Monostearate, Type II; Cremer Oleo GmbH&Co. KG company (Hamburg, Germany); Kolliwax^®^ GMS II, Glycerol Monostearate 40,55 Type II; BASF (Hamburg, Germany); Precirol^®^ ATO 5, Glycerol Distearate Type I EP; Gattefossé (Madrid, Spain); Witepsol^®^ E85 (CREMER OLEO GmbH & Co. KG, Hamburg, Germany), Hard Fat, Adeps solidus; Cremer Oleo GmbH&Co. KG company (Hamburg, Germany) were selected as solid lipids for screening the solubility of alpha-pinene. Poloxamer 188 (trade name: Kolliphor^®^ P188) was used as surfactant. Poloxamer 188, bought from the company BASF (Ludwigshafen, Germany), is a non-ionic triblock copolymer composed of central hydrophobic polyoxypropylene (poly(propylene oxide) PPOx, where x = 28) chain surrounded by two hydrophilic chains of polyoxyethylene (poly(ethylene oxide, PEOy, where y = 79). Alpha-pinene (C10H16), also known as (1R,5R)-2-Pinene or (1R,5R)-2,6,6-Trimethylbicyclo[3.1.1]hept-2-ene), was purchased from Sigma Aldrich (Madrid, Spain). Ultra-purified water was obtained from a Milli-Q^®^ Plus system (Millipore, Germany) and filtered through a 0.22 μm nylon filter before use. All reagents were analytically pure and were used without further treatment.

### 2.2. Lipid Screening

To produce SLN, the active compound is added to the melted lipid; therefore, it is necessary to select a lipid that solubilizes the active compound completely. The appropriate lipid screening relays on the solubility of the active compound in an optimal ratio of solid lipid to drug in order to obtain a visually clear solution in the lipid melt under normal light during naked eye observation [5].

In this work, 8 different solid lipids—Compritol^®^ 888 ATO, Dynasan^®^ 116, Dynasan^®^ 118, Dynasan^®^ P60 (F), Imwitor^®^ 900K, Kolliwax^®^ GMS II, Precirol^®^ ATO 5 and Witepsol^®^ E85—were selected for the screening approach. Alpha-pinene, as an active compound, was added to each of 15 mL glass vials containing the same quantity of the selected lipid in the ratio 1:100. Then, all of mixtures in vials were heated above the melting point of each lipid in the controlled temperature oven (80 °C up to 1 h). The solubility of the active compound was observed after 15 min, 30 min, and 1 h under 80 °C, as well as after 24 h and 72 h of storage at room temperature (25 °C) since the solidification of the lipid and active compound mixture.

Another way to perform a preliminary lipid screening is to carry out microscopic observation of the mixture of selected lipid with an active compound, after mixing them with water. Vigorous mixing that occurs at the water–formulation interface is usually accompanied by diffusion and stranding mechanisms, which may indicate an efficient emulsification. Moreover, the absence of active substance precipitate after complete mixing of the formulation with aqueous medium is also a requirement [45].

### 2.3. Experimental Factorial Design

The identification of the influencing parameters that will affect the final dosage form is of significance in the design of a new pharmaceutical formulation. The experimental factorial design helps to analyze the influence of the different independent variables on the properties of the drug delivery system. This statistical analysis also provides a means for the selection of the most optimal experimental conditions, such as different ratios of surfactants and different amounts of lipids. Experimental factorial design is an effective statistical approach to estimate the influence of independent variables on the dependent variables (in this study: mean particle size (Z-Ave), polydispersity index (PDI) and zeta potential (ZP)), which ultimately determine the physicochemical properties of lipid nanoparticles.

A factorial design approach was applied to maximize the experimental efficiency requiring a minimum of experiments to optimize the SLN production [46]. The influence of the surfactant ratio (Poloxamer 188) and lipid ratio (Imwitor^®^ 900 K) on α-pinene-loaded SLN was evaluated by using a 2^2^ factorial design composed of 2 variables, which were set at 2-levels each. The dependent variables were mean particle size (Z-Ave), polydispersity index (PDI) and zeta potential (ZP). For each factor, the lower and higher values of the lower and upper levels were represented by (−1) and (+1), respectively. The central point was replicated three times for estimating the experimental error represented by (0) (Table 1). The levels were chosen on the basis of the tested lower and upper values for each variable, according to pre-formulation studies as well as literature research [47]. The data were analyzed by STATISTICA 7.0^®^ (Stafsoft, Inc., Tulsa, OK, USA) software. The SLN dispersions were randomly produced. In order to identify the significance of the effects and interactions between them, an analysis of variance statistical test (ANOVA, San Francisco, CA, USA) was performed for each response parameter. A *p*-value < 0.1 was considered statistically significant.

### 2.4. Preparation of Nanoparticles

All of the SLN dispersions were produced by dispersing the lipid phase, composed of α-pinene (1 wt.%) and Imwitor^®^ 900K (2, 4 or 8 wt.%) at the same temperature (both at 5–10 °C above the melting point of lipid), in an aqueous solution of Poloxamer 188 (1.25, 2.5 or 5 wt.%) using the hot high pressure homogenization technique. Briefly, the pre-emulsion was processed in a high-shear mixing Ultra-Turrax^®^ T25 Digital Homogenizer (Ystral GmbH D-7801, Dottingen, Germany) at 10,000 rpm for 10 min followed by hot HPH (Emulsiflex-C3, Avestin, Inc., Ottawa, ON, Canada) for 30 min under a pressure of 300 bar. As a result, 7 SLN dispersions were produced.

### 2.5. Characterization of Nanoparticles

Dynamic light scattering (DLS) was employed to record the variation in the intensity of the scattered light on the microsecond time scale. The measuring principle of DLS was based on particles in gas or liquid being subjected to Brownian motions. The movement (diffusion) of the particles can be described by the Stokes-Einstein equation [36]. Z-average (Z-Ave) size or average particle size Z (also called the average cumulative) and polydispersity index (PDI) were determined using Zetasizer Nano ZS (Malvern, Worcestershire, UK), which was equipped with a particle size range of 0.3 nm to 10 μm and a laser beam (λ = 633 nm; 4 mW). A scattered light detector was positioned at an angle of 173° (non-invasive backscatter) in order to unmask scattered light signals of low intensity originating from smaller particles. In this study, analyzed samples were diluted 100 times in ultra-purified water. The Z-Ave and PDI were measured in triplicate during one cycle (one cycle corresponding to 10 runs) from three independent samplings of the same batch. Data were then expressed as an arithmetical means ± standard deviations (SD).

Zeta potential (ZP), as an electrokinetic potential in colloidal dispersions, can exist at the slipping plane (the boundary of the electrical double layer of the particle). Zeta potential can also be determined as the potential difference between the dispersion medium and the stationary layer of fluid attached to the dispersed particle. The value of ZP usually indicates the degree of electrostatic repulsion between adjacent, similarly charged particles in a dispersion [36]. In this work, the measurements of ZP were performed using Zetasizer Nano ZS (Malvern, Worcestershire, UK). Analyzed samples were diluted 100 times in ultra-purified water and the value of ZP was measured triplicate during one cycle (30 runs in each measurement). Afterwards, data were expressed as the arithmetical means and SD could be calculated. The measuring principle was based on a technique ELS (electrophoretic light scattering), whereas the electrophoretic mobility was obtained by performing an electrophoresis experiment on the sample. As a result, the velocity of the particles using Laser Doppler Velocimetry (LDV) could also be measured [36]. The zeta potential was calculated using the Helmholtz-Smoluchowski equation that was included in the software system. The obtained values have been presented as the mean of triplicate runs per sample, including standard deviations.

### 2.6. Accelerated Stability Analysis

Dispersion analyzer, known as LUMiSizer^®^ (Boulder, CO, USA), has been helpful for a quick characterization of any demixing phenomena, like sedimentation, flotation or consolidation. This multi-sample analytical centrifuge has allowed for the calculation of the velocity distribution in the centrifugal field as well as of particle size distribution [48,49]. By using LUMiSizer^®^ (Boulder, CO, USA), the variation of transmitted light over time and space have been recorded in transmission profiles. It can provide the information on the separation process kinetics, as well allowing the calculation of particle migration velocity, which is intimately related to the particle size distribution [43,44,49]. Moreover, by using LUMiSizer^®^ (Boulder, CO, USA), a fast stability ranking and shelf-life estimation of undiluted dispersions at their original concentration (in minutes/hours instead of months/years) has been provided [47]. This centrifugal sedimentation method has also employed the STEP-Technology^®^, permitting to obtain space- and time-resolved extinction profiles, while the shape and progression of the transmission profiles have contained the information on the kinetics of the separation process, facilitating particle characterization. The separation behavior for each sample has been compared and carefully analyzed by tracing the variation in transmission at any part of the sample or by tracing the movement of any phase boundary. Based on the extinction profiles, instability processes have been quantified regarding clarification velocity, sedimentation and flotation velocity of particles, residual turbidity, and separated phase volume (liquid or solid) [47,50]. The evolution of the transmission profiles of tested samples has enabled the analysis of their demixing behavior and stability. In turn, stable colloidal dispersions have allowed the formation of a flatbed under a centrifugal field [47].

In this work, LUMiSizer^®^ (Boulder, CO, USA) has recorded the evolution of the transmission profiles and trace instability phenomena using the SEPView^®^ software. The software has also enabled the comparison of different samples and give information about the instability index of each sample. In this case, the amount of 0.5 mL of SLN dispersions were placed in the cell, subjected to 4000 rpm rotor speed at 25 °C. A total of 750 profiles were recorded in intervals of 20 s. Instability indexes were calculated by the SEPView^®^ software.

## 3. Results and Discussion

### 3.1. Lipid Screening

Table 2 shows the results of the lipid screening carried out over time for a set of raw materials. Two solid lipids (Imwitor^®^ 900 K and Compritol^®^ 888 ATO) demonstrated to solubilize α-pinene over the time course of 72 h. Imwitor^®^ 900 K has been selected for further experiments, firstly due to its long alkyl chain; secondly, because of amphiphilic character of the structural formula of this glycerol monostearate. For these two reasons, a good stability for all of polar and non-polar compounds can be provided. Furthermore, α-pinene is an example of non-polar compound from the groups of hydrocarbons; therefore, it can be easily soluble in the compounds having the alkyl chains. This justifies why Compritol^®^ 888 ATO (glyceryl dibehenate) might also be a suitable solid lipid for the production of α-pinene-loaded SLN, as shown in Table 2. Additionally, Imwitor^®^ 900 K has a lower melting point (≈61 °C) than Compritol^®^ 888 ATO (≈70 °C) which favors the loading of volatile compounds.

### 3.2. Experimental Factorial Design

As main shortcoming of experimental design is to deal with increasing number of factors and/or levels, in this work 2^2^ full factorial design was developed in order to optimize α-pinene-loaded SLN. In this case, seven different formulations were produced with different ratios of lipid (Imwitor^®^ 900K) and surfactant (Poloxamer 188). SLN were stored at room temperature (25 °C). The mean particle size, polydispersity index and zeta potential were measured 24 h after the production. The obtained results are shown in Table 3. The mean particle size varied from 136.7 ± 0.7 nm (SLN 5) to 3002.3 ± 268.88 nm (SLN2), whereas the PDI ranged from 0.170 ± 0.01 (SLN5) to 0.775 ± 0.29 (SLN2). Zeta potential was approximately 0 mV in all formulations since the surfactant has owned a non-ionic nature that might cause a formation of a spherically stabilizing adsorbed polymer layer in the SLN surface [51].

For each three dependent variables, analysis of the variance (ANOVA) was performed using a confidence level of 90% (*p*-value = 0.1), because the results with the most used 95% confidence interval (*p*-value = 0.05) will not be statistically significant.

The statistical significance (*p*-value < 0.1) of the different ratios for both of Imwitor^®^ 900K and Poloxamer 188, as well as their interaction on the Z-Ave are shown in Table 4. Additionally, neither one of the single independent variables, nor the interaction between PDI and ZP had a significant effect (*p*-value > 0.1), as also shown in Appendix A.

The response coefficients were studied for their statistical significance by Pareto chart. These results are shown in Figure 2. The Pareto charts set the *t*-value of effect. The variation of the low value to a high value of Poloxamer 188 concentration had a negative effect on the particle size (*t*-value = −2.53267; Figure 2A). Similarly, the interaction between the variation of the surfactant and lipid concentration from lower to higher values had a significant negative effect on the particle size (*t*-value = −2.43725). On the other hand, the concentration of solid lipid had a prevailing positive effect on the particle size (*t*-value = 2.484696). Likewise, the surfactant concentration and the interaction between factors had a negative effect on the PDI (Figure 2B), whereas the lipid concentration had a positive effect.

The interactive effects of the different dependent variables were plotted in three-dimensional (3D) response surface charts. In the surface response charts (Figure 3), the variations in the response values were plotted in the *Z*-axis against the levels of the two independent variables (Poloxamer 188 in *X*-axis and Imwitor^®^ 900 K in *Y*-axis). The increase of the surfactant concentration decreased the particle size to values below 500 nm (Figure 3A), while the increasing of the lipid concentration effected on the particle size (up to values above the nanometric range). Based on scientific reports, it may be concluded that the increase of the solid lipid concentration will also increase the viscosity of the system. Therefore, the surface tension has been enhanced and the particle agglomeration may affect the mean particle size [46]. The increase of the surfactant concentration decreased the PDI down to values lower than 0.4.

As shown in Table 3, the ZP values were kept at 0 mV in all tested experiments and indicated that the independent variables (lipid and surfactant concentrations), as well as their interaction in the two levels tested, had no influence on the electrical charge of the α-pinene-loaded SLN (Figure 2C). Such low ZP values were attributed to the presence of the hydrophilic chains of poloxamer 188 (polyoxypropylene–polyoxyethylene co-polymer) which also contributed to improving the colloidal stability of SLN in dispersion, limiting the risk of aggregation.

Formulation SLN5 was selected as the best formulation attributed to its lower particle size parameters, i.e., mean particle size of 136.7 nm and PDI value of 0.170. Based on the obtained results, the optimal SLN dispersion was the central point composed of 1 wt.% α-pinene, 4 wt.% of Imwitor^®^ 900K and 2.5 wt.% of Poloxamer 188.

### 3.3. Stability Analysis with LUMiSizer^®^

A centrifugal separation analysis (CSA) method, LUMiSizer^®^, was used in order to assess the stability of the produced α-pinene-SLN dispersions. A fast and accurate characterization of the optimized α-pinene-loaded SLN (SLN5) was carried out and the recorded profiles are presented in Figure 4. Results of remaining SLN formulations are provided in Appendix A.

Data shown in Figure 4 and in Appendix A demonstrate that sedimentation was coupled to flocculation for SLN1, SLN3, SLN5, SLN6 and SLN7. SLN dispersions can translate heterogeneous particle size distribution, when the particles migrate with different velocities. On the other hand, due to a variation in the light transmission recorded for SLN2 and SLN4, a “flat bed profile” typical of a stable formulation [50] has been described for these latter.

Besides the qualitative information about the type of separation process that occurs, the use of LUMiSizer^®^ has not only allowed to obtain the quantitative data, but also to calculate the instability index. This analysis could be performed based on the clarification (increase in transmission due to phase separation by sedimentation) at a given separation time, divided by the maximum clarification. The results containing all of seven produced α-pinene-loaded SLN stored at 25 °C are shown in Figure 5. The instability index is a dimensionless number and ranges from 0 (more stable) to 1 (more unstable). This means that, for the same total clarification, samples with high clarification rates tend to be more unstable [47]. An instability index has been essential to detect a potentially appearing separation process. It should be highlighted that if the separation occurs, the instability index can be an indicator of the speed of this process. SLN1 has depicted the highest instability index, confirming the results obtained with the separation profiles (Figure 4). SLN2 and SLN4 have shown to be the most stable ones, while the samples SLN3, SLN5 and SLN7 have reached an intermediate instability index value.

Based on the centrifugation process and assuming a spherical shape for the particles, there is a quadratic relation between sedimentation velocity and the particle radius: $k\cdot r^{2}=V_{m}$, in which *k* is a constant, *r* is the radius of the particle and *V_m_* is the sedimentation velocity. According to the LUMiSizer^®^ analysis, SLN2 and SLN4 formulations have yielded very similar populations, which showed no visible separation during the time of the experiment. Consequently, it was not possible to calculate any value for sedimentation velocity. These particles were very small and led to a stable dispersion, as confirmed by the kinetic profiles and instability index. On the other hand, SLN1 formulation population had a higher particle size, which translated to a higher migration velocity and lower stability. Results are summarized in Table 4.

Based on the obtained results, empty-SLN and α-pinene-loaded-SLN, were prepared for the centrifugal separation reanalysis. The empty-SLN dispersion was composed of 5 wt.% of Imwitor^®^ 900K and 2.5 wt.% of Poloxamer 188, while the α-pinene-SLN dispersion was enriched with the drug being composed of 1 wt.% α-pinene, 4 wt.% of Imwitor^®^ 900K and 2.5 wt.% of Poloxamer 188. In order to evaluate how temperature and light affect the stability of the produced α-pinene-SLN dispersions [52], these were stored at three different temperatures (4, 25 and 40 °C) for 1 month followed by analysis with LUMiSizer^®^. The transmission profiles of tested SLN have been shown in Figure 6.

Data presented in Figure 6 have shown that the creaming process has occurred over time in all tested formulations. With the increase of the temperature, both the empty and α-pinene-loaded SLN tend to become more unstable, with the greatest changes more evident at 40 °C. This has indicated that the high temperature may significantly increase the instability, leading to the occurrence of fast instability phenomena, such as creaming and sedimentation.

In the case of α-pinene-loaded SLN, step profiles were not significant different between the samples that were stored at 4 and 25 °C. This could suggest that the physical stability of the dispersion cannot be greatly influenced in this range of temperature. In the case of α-pinene SLN stored at 40 °C, an increase in separation profiles was observed, indicating a higher degree of instability. It has been further confirmed by the instability index (0.163) and the velocity value in D90%.

The α-pinene-loaded SLN has shown a lower light transmission and a more stable kinetic profile than the empty-SLN. In the empty-SLN, particles migrate to the top of the cell with different velocities in all tested temperatures. The results may suggest that α-pinene can tend to stabilize the particles, because it is able to induce a nanostructure of the matrix, altering the polymorphic form of the lipid.

By applying LUMiSizer^®^ technique, the comparison of instability index for empty-SLN and α-pinene-loaded SLN stored at 4, 25 and 40 °C could also be performed. The results have been shown in Figure 7. In this regard, the highest values of instability index (0.151–0.584) were observed for the empty-SLN stored at all of different temperatures (4, 25, and 40 °C). On the other hand, the formulations of α-pinene-loaded SLN stored at 4, 25 and 40 °C showed that the instability index ranged from 0.039 to 0.163.

The α-pinene-SLN stored at 4 and 25 °C has gained the lowest instability index, confirming the results that were obtained with the separation profile, in which a slow separation process was observed. Results have clearly indicated that the increase of the temperature can lead to a decrease in the stability. This process has been observed for both of the empty as well as α-pinene-loaded SLN. The obtained results have attributed to the increase in the temperature. It leads to a decrease in the medium viscosity and makes the movement of particles easier. Therefore, a faster separation velocity can be expected when it is compared to lower temperatures [53,54].

According to the LUMiSizer^®^ analysis, the sedimentation velocities have indicated that α-pinene-loaded SLN stored at 4 and 25 °C were very small, providing a stable dispersion and confirming the previous results of the instability index. The short range between D10% and D90% have translated to a more unimodal particle size distribution. In the case of α-pinene-loaded SLN stored at 40 °C, a very wide distribution in the particle size range was recorded. The velocity of sedimentation ranged from 1.57 µm/s to 296.8 µm/s. A wide range of sedimentation speed has also been observed for the empty-SLN, suggesting that a high temperature may be related with this observation. Empty-SLN stored at 4 and 25 °C have shown a wider range in sedimentation velocity in the comparison with the formulations loaded with α-pinene. It has led to more unstable suspensions, corroborating the previous results obtained (STEP profiles and instability index). Results are summarized in Table 5.

## 4. Conclusions

The main purpose of the experimental factorial design was the analysis of the effects for different factors with the simultaneously determination of the factors’ interaction. In pharmaceutical technology, a factorial design can be commonly used to find the optimal drug delivery systems by performing a minimum of experiments. The influence of independent variables, such as concentration of solid lipid and surfactant, on the dependent variables (Z-Ave, PDI, and ZP) was effectively assessed by statistical analysis. The results have proven that the central point (0) was the most optimal of all prepared α-pinene-loaded SLN formulation. The SLN sample was composed of 1 wt.% of α-pinene, 4 wt.% of Imwitor^®^ 900K and 2 wt.% of Poloxamer188 and was replicated in triplicate. As a result, SLN5, with 136.7 nm of Z-Ave, 0.170 of PDI and |±0 mV| of ZP, was selected for the stability assessment. We have demonstrated that LUMiSizer^®^ can be successfully used in the kinetic analysis of α-pinene-SLN during accelerated storage time. Alpha-pinene-loaded SLN exhibited a higher stability when stored at 4 and 25 °C (with instability index of 0.039 and 0.069, respectively), compared to the same formulation stored at 40 °C (instability index of 0.163).

## Figures and Tables

**Figure 1 molecules-24-02683-f001:**
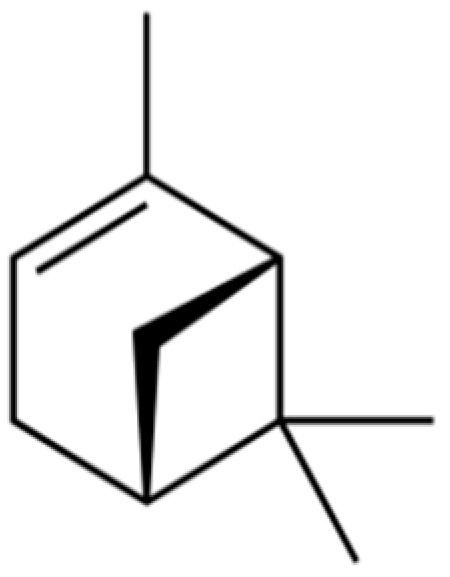
Structural formula of the (1R)-(+)-α-pinene.

**Figure 2 molecules-24-02683-f002:**
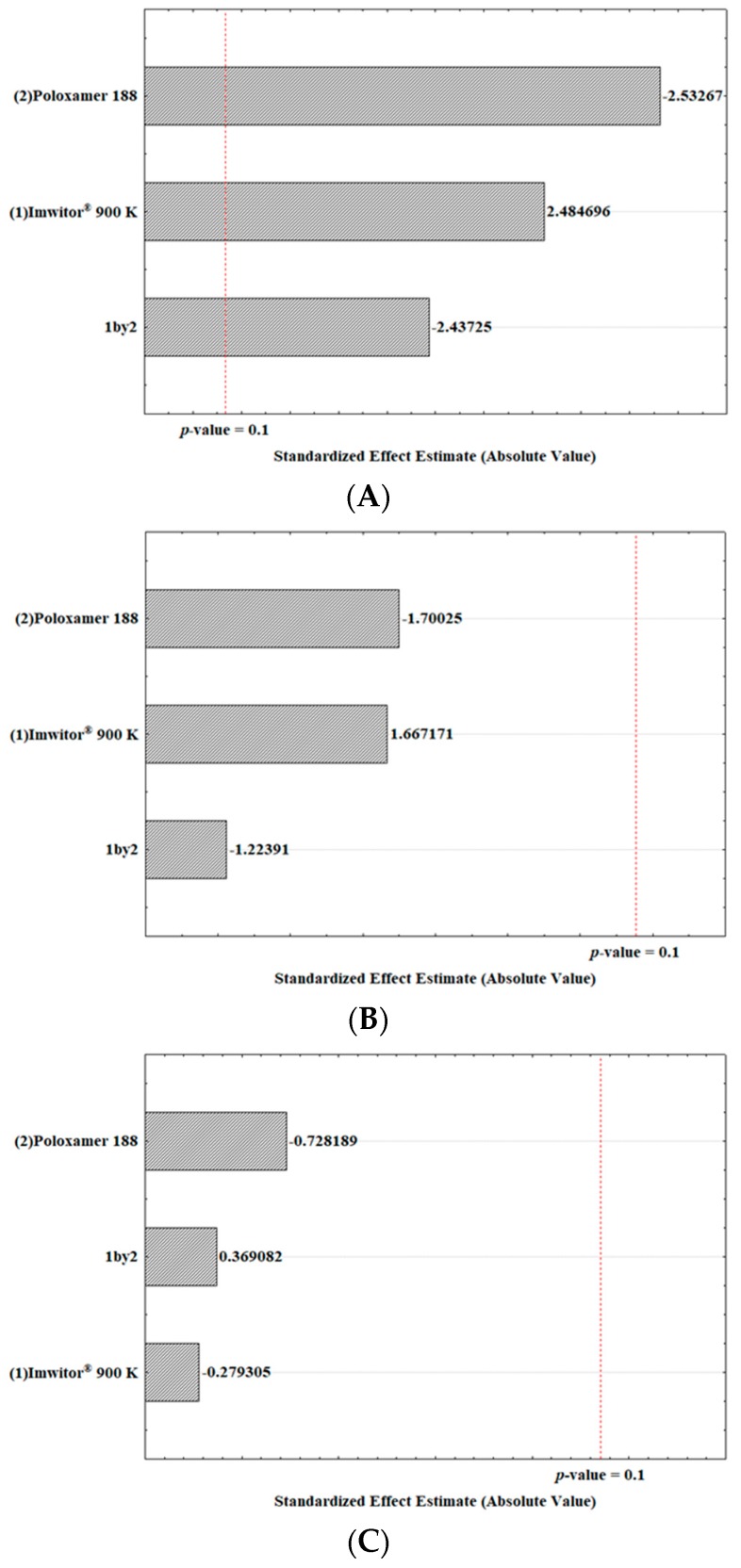
Pareto charts of the analyzed effects of the concentration variation of the Imwitor^®^ 900 K as a solid lipid (1), Poloxamer 188 as a surfactant (2) and the interaction of both (1 by 2) for the Z-Ave (**A**), PDI (**B**) and ZP (**C**).

**Figure 3 molecules-24-02683-f003:**
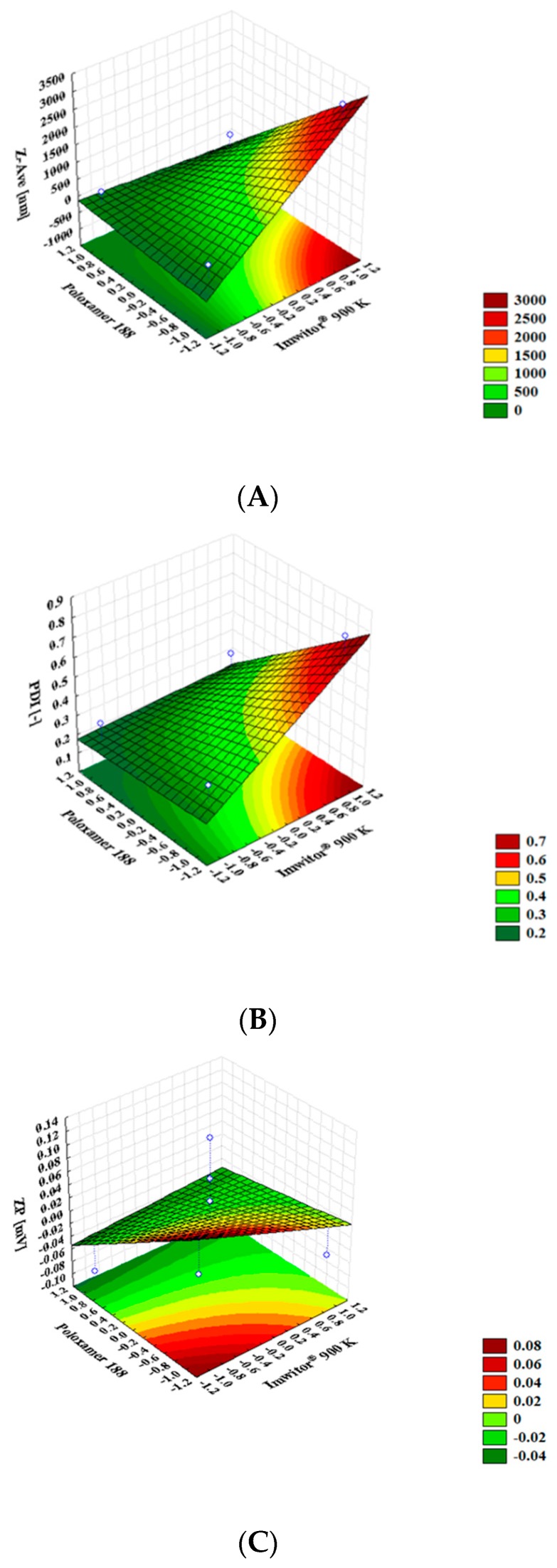
Surface response charts of the influence of the wt. % of Imwitor^®^ 900K and Poloxamer 188 on the Z-Ave (**A**), on the PDI (**B**) and on ZP (**C**).

**Figure 4 molecules-24-02683-f004:**
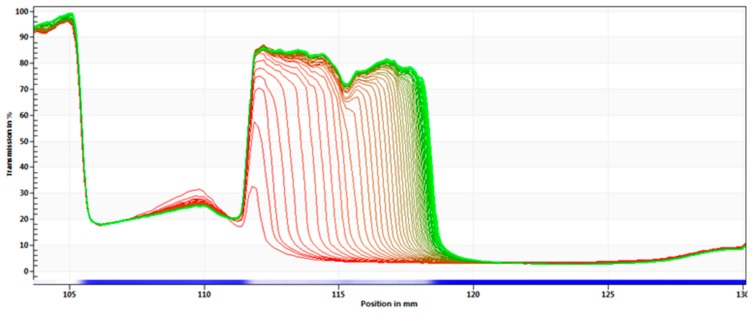
Space- and time-resolved extinction profiles (STEP) with analysis of phenomena observed for the optimized α-pinene-loaded SLN (SLN5) stored at 25 °C, showing separation process: sedimentation with particle compaction.

**Figure 5 molecules-24-02683-f005:**
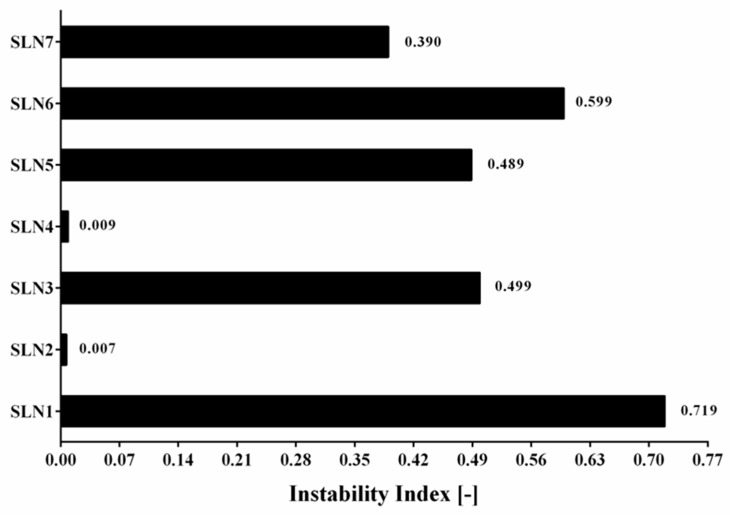
Graphical comparison of instability index for all of 7 produced α-pinene-loaded SLN stored at 25 °C.

**Figure 6 molecules-24-02683-f006:**
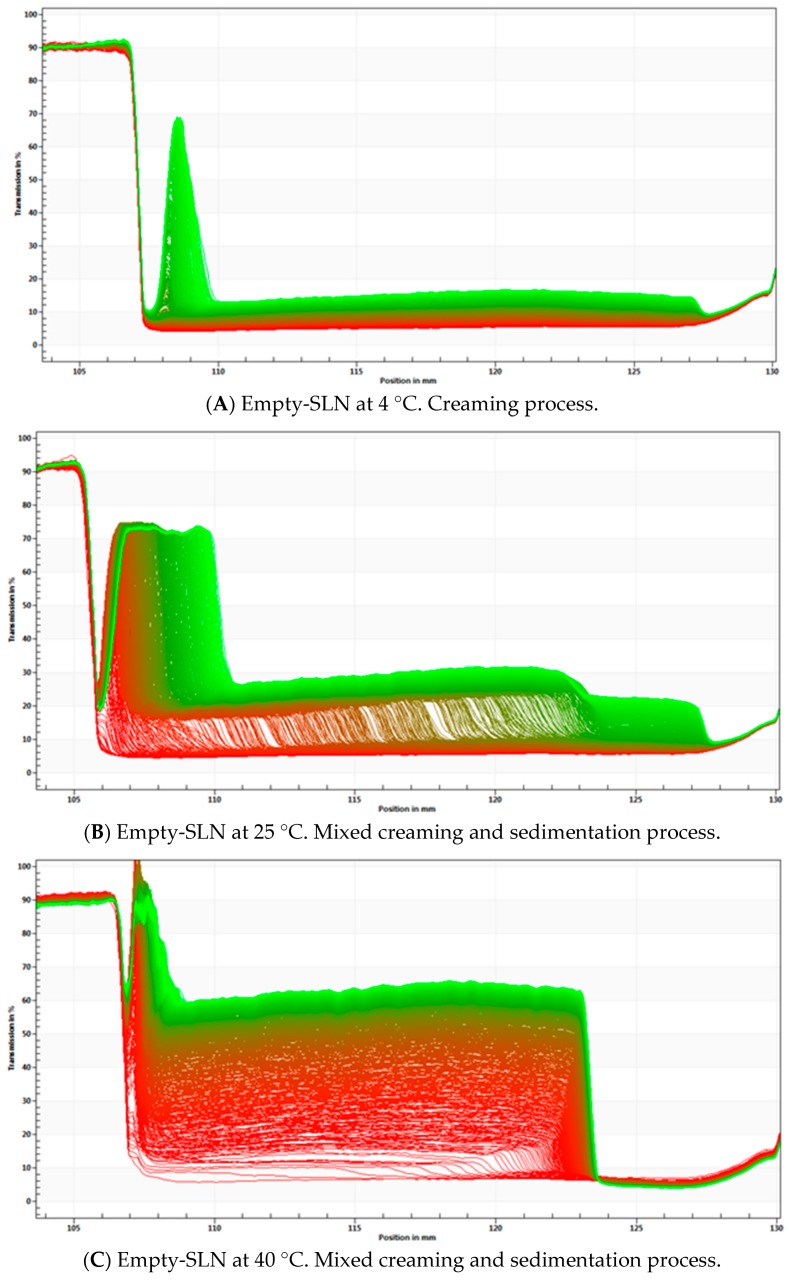
STEP Profiles with analysis of phenomena observed for empty-SLN at 4 °C (**A**), 25 °C (**B**), 40 °C (**C**) and α-pinene-loaded SLN stored at 4 °C (**D**), 25 °C (**E**) and 40 °C (**F**).

**Figure 7 molecules-24-02683-f007:**
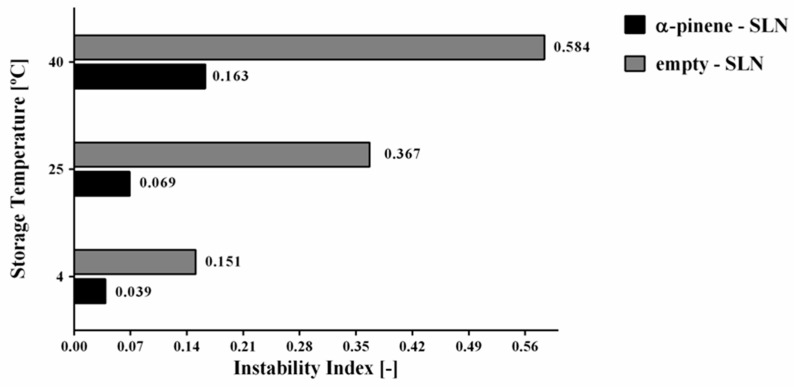
Graphical comparison of instability index for empty-SLN and α-pinene-loaded SLN stored at 4, 25 and 40 °C.

**Table 1 molecules-24-02683-t001:** Initial 2-level factorial design, providing the lower (−1), upper (+1) levels and central point (0) values for each studied variable.

Variables	Levels		
	Low (−1)	Central Point (0)	High (+1)
**Imwitor^®^ 900 K (wt.%)**	2	4	8
**Poloxamer 188 (wt.%)**	1.25	2.5	5

**Table 2 molecules-24-02683-t002:** Tested lipids containing α-pinene in the ratio 1:100, observed over time.

Name of Lipid + α-Pinene (Ratio 1:100)	Solubility (Naked Eye)				
	15 min.	30 min.	1 h	24 h	72 h
**Compritol^®^ 888 ATO**	☑	☑	☑	☑	☑
**Dynasan^®^ 116**	⊠	⊠	⊠	⊠	⊠
**Dynasan^®^118**	⊠	⊠	⊠	⊠	⊠
**Dynasan^®^ P 60 (F)**	☑	☑	☑	☑	⊠
**Imwitor^®^ 900 K**	☑	☑	☑	☑	☑
**Kolliwax^®^ GMS II**	⊠	⊠	⊠	⊠	⊠
**Precirol^®^ ATO 5**	⊠	⊠	⊠	⊠	⊠
**Witepsol^®^ E85**	☑	☑	☑	☑	⊠

⊠ - insoluble; ☑ - soluble.

**Table 3 molecules-24-02683-t003:** Response dependent variables (Z-Ave, PDI and ZP) of the two independent factors presented in Table 1 for all of 7 produced α-pinene-loaded SLN.

	Independent Variables		Dependent Variables		
Sample Name	Imwitor^®^ 900 K (wt.%)	Poloxamer 188 (wt.%)	Z-Ave (nm) ± SD	PDI (–) ± SD	ZP (mV) ± SD
SLN1	2	1.25	211.6 ± 2.0	0.338 ± 0.020	0.016 ± 0.110
SLN2	8	1.25	3002.3 ± 268.9	0.775 ± 0.290	−0.049 ± 0.100
SLN3	2	5	157.5 ± 0.8	0.266 ± 0.000	−0.094 ± 0.190
SLN4	8	5	184.4 ± 0.9	0.333 ± 0.010	−0.085 ± 0.090
SLN5	4	2.5	136.7 ± 0.7	0.170 ± 0.010	0.060 ± 0.170
SLN6	4	2.5	142.7 ± 1.2	0.276 ± 0.010	0.121 ± 0.120
SLN7	4	2.5	137.3 ± 1.0	0.270 ± 0.010	0.026 ± 0.350

Z-Ave—mean particle size; PDI—polydispersity index; ZP—zeta potential.

**Table 4 molecules-24-02683-t004:** Distribution (D) of particles sedimentation (velocity) for all of 7 produced α-pinene-loaded SLN.

Sample Name	D10% (µm/s)	D50% (µm/s)	D90% (µm/s)
**SLN1**	137.1	180.9	220.3
**SLN2**	*	*	*
**SLN3**	1.123	47.02	548.2
**SLN4**	*	*	*
**SLN5**	42.67	75.35	133.3
**SLN6**	36.90	66.97	96.23
**SLN7**	30.06	72.96	189.0

* Velocity of sedimentation not determined as no separation was visible. Note: All samples (SLN1–SLN7) were stored at 25 °C.

**Table 5 molecules-24-02683-t005:** Distribution (D) of particles sedimentation (velocity) for empty-SLN and α-pinene-SLN stored at 4, 25 and 40 °C.

Sample Name	Storage Temperature (°C)	D10% (µm/s)	D50% (µm/s)	D90% (µm/s)
**empty-SLN**	4	0.9722	1.919	12.49
**empty-SLN**	25	1.989	11.79	62.20
**empty-SLN**	40	0.2720	2.283	55.05
**α-pinene-SLN**	4	1.413	2.102	4.732
**α-pinene-SLN**	25	1.356	2.270	4.555
**α-pinene-SLN**	40	1.570	8.535	296.8

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
