# Peer review of "Development and Optimization of Alpha-Pinene-Loaded Solid Lipid Nanoparticles (SLN) Using Experimental Factorial Design and Dispersion Analysis"

_molecules, 2019, doi:10.3390/molecules24152683_

Round 1

Reviewer 1 Report

This article presents the encapsulation of bicyclic monoterpene α-pinene into solid lipid nanoparticles, and has interesting results using experimental factorial design, but needs some corrections such as:

All the figures, as they are with the axes with low visibility and low quality. You should increase the font.should correct in the summary (line 37,...polydispersity index (PDI) and zeta potential (ZP), was estimated using a 22)

Author Response

The authors acknowledge the referee’s comments, which have been taken into acount in the revised version. Figures have been obtained from the system’s software and cannot be edited. We have however increased the size as requested by the referee. Text size has also been increased whenever applicable. Typos and use of English have been thoroughly revised. Editing is marked in blue.

Reviewer 2 Report

Zielińska et al in their manuscript ‘Development and Optimization of alpha-pinene-loaded solid lipid nanoparticles (SLN) using experimental design and dispersion analysis’ have characterised the SLNs using LUMISizer. Congratulations on bringing together a well-written manuscript. In my opinion, this is a good manuscript and the use of LumiSizer for stability characterization will aid other researcers to look for similar approaches. Having said that, I have a few suggestions that may improve the quality of the manuscript further.

Major comments:

1.       Methods section suggests that the measurements of PS, ZP were done in triplicates. However, were the formulations also prepared in triplicates? From Table 3, it is clear that SLN5, 6, and 7 are three replicates, what about others? Is this a disadvantage of using this experimental design?

2.       The cargo used here is alpha-pinene. In my opinion, all constitutents of essential oils are also volatile like the parent essential oil. As such, when you use high temperature homogenisation method, would alpha-pinene volatilize? Please include encapsulation efficiency results to show that alpha-pinene is encapsulated. DSC, XRD results would also work.

Minor comments:

1.       Editorial – there is no space between number and °C for example 25°C and not 25 °C. Also, please correct º to °.

2.       Section 2.5: Characterization of nanoparticles. Please change the heading.

3.       Section 3.1: Authors have mentioned that Compritol 888 ATO is also suitable but have not shown any results. Was this just a suggestion or a direction that the authors have taken but not presented here? Please provide a clarification in the manuscript.

4.       Table 3: Use 1 sig fig for Z-ave, 2 sig fig for PDI and ZP.

5.       Table 4: I think this can go in the Supplementary section.

6.       Line 303: The authors say the ZP was kept at 0 mV. How was that achieved? Seems very difficult, because that would cause the particles to sediment/flocculate. Please explain/

7.       Figure 4. A – should that be SLN1? Please correct the title. Also, one figure illustrating particle sedimentation and one illustrating no sedimentation is enough. Others can be moved to supplementary section. Alternatively, can these be overlayed and shown in different colours?

8.       Instability index – do you have a formula? If so please include (replace lines 375-377 with equation). Provide reference too. This is a very good point and may be helpful for other researchers.

9.       Similarly, line 392, there is an equation. Provide equation numbers to all equations in the manuscript and if possible have them on a separate line.

10.   Table 5: Errors? Or standard deviations should be included.

11.   Figure 6: Overlay all figures.

Author Response

Review Report Form

Open Review

English language and style

( ) Extensive editing of English language and style required 
( ) Moderate English changes required 
(x) English language and style are fine/minor spell check required 
( ) I don't feel qualified to judge about the English language and style 

Yes

Can be improved

Must be improved

Not applicable

Does the   introduction provide sufficient background and include all relevant   references?

(x)

( )

( )

( )

Is the research   design appropriate?

( )

(x)

( )

( )

Are the methods   adequately described?

(x)

( )

( )

( )

Are the results   clearly presented?

( )

(x)

( )

( )

Are the   conclusions supported by the results?

( )

(x)

( )

( )

Comments and Suggestions for Authors

Zielińska et al in their manuscript ‘Development and Optimization of alpha-pinene-loaded solid lipid nanoparticles (SLN) using experimental design and dispersion analysis’ have characterised the SLNs using LUMISizer. Congratulations on bringing together a well-written manuscript. In my opinion, this is a good manuscript and the use of LumiSizer for stability characterization will aid other researcers to look for similar approaches. Having said that, I have a few suggestions that may improve the quality of the manuscript further.

Major comments:

1.       Methods section suggests that the measurements of PS, ZP were done in triplicates. However, were the formulations also prepared in triplicates? From Table 3, it is clear that SLN5, 6, and 7 are three replicates, what about others? Is this a disadvantage of using this experimental design?

We acknowledge the referee’s remark. The formulations have not been prepared in triplicate, but for all formulations (SLN1-SLN7) the particle size and ZP have been measured trice from independent samplings of the same batch. Each measurement is run 10-times which means that reported results are from 30 runs. To clarify this issue, in section 2.5 we have edited the sentence as follows: “The Z-Ave and PDI were measured in triplicate during one cycle (one cycle corresponding to 10 runs) from three independent samplings of the same batch. Data were then expressed as an arithmetical means ± standard deviations (SD).”. SLN 5, 6 and 7 are three independent formulations produced from the very same quantitative and qualitative composition. The experimental design reported in this work has been a 22 factorial design composed of 2 variables, which were set at 2-levels each. As stated in the manuscript, the central point was replicated in triplicate, which means that 3 independent batches of the same formulation have been produced and analised. These are represented by SLN 5, 6 and 7. The main disadvantage of experimental design is to deal with increasing number of factors and/or levels. To limit this, we have used only two variables set at two levels each (-1) and (+1). To address this query, in section 3.2 we have also edited the sentence as follows: “As main shortcoming of experimental design is to deal with increasing number of factors and/or levels, in this work 22 full factorial design was developed in order to optimize α-pinene-loaded SLN.”.

2.       The cargo used here is alpha-pinene. In my opinion, all constitutents of essential oils are also volatile like the parent essential oil. As such, when you use high temperature homogenisation method, would alpha-pinene volatilize? Please include encapsulation efficiency results to show that alpha-pinene is encapsulated. DSC, XRD results would also work.

We acknowledge the referee’s remark. Alpha-pinene is indeed volatile and the data that are being asked by the referee have been recorded and are being considered for publication (entitled: “Release kinetics, loading and stability assessment of monoterpenes-loaded Solid Lipid Nanoparticles (SLN)”) in a different journal. The results reported here in the present work are focused on the selection of the best lipid for the loading of this active ingredient (cargo) and respective stability of the optimized formulations obtained from the factorial design. This work is already extensive (6 tables, 7 figures). We, nevertheless, anticipate that release kinetics, encapsulation efficiency, DSC and XRD profiles were found to be very much dependent on the type of monoterpenes.

Minor comments:

1.       Editorial – there is no space between number and °C for example 25°C and not 25 °C. Also, please correct º to °.

We have proceeded accordingly. Corrections have been made in all cases.

2.       Section 2.5: Characterization of nanoparticles. Please change the heading.

Heading has been corrected.

3.       Section 3.1: Authors have mentioned that Compritol 888 ATO is also suitable but have not shown any results. Was this just a suggestion or a direction that the authors have taken but not presented here? Please provide a clarification in the manuscript.

According to the results show in Table 2, compritol was also shown to dissolve the active over the time course of the experiment. We have included the following explanation:Table 2 shows the results of the lipid screening carried out over time for a set of raw materials. Two solid lipids (Imwitor® 900 K and Compritol® 888 ATO) demonstrated to solubilize α-pinene over the time course of 72 hours. Imwitor® 900 K has been selected for further experiments, firstly due to its long alkyl chain; secondly, because of amphiphilic character of the structural formula of this glycerol monostearate. For these two reasons, a good stability for all of polar and non-polar compounds can be provided. Furthermore, α-pinene is an example of non-polar compound from the groups of hydrocarbons, therefore can be easily soluble in the compounds having the alkyl chains. It justifies why Compritol® 888 ATO (glyceryl dibehenate) might also be a suitable solid lipid for the production of α-pinene-loaded SLN, as shown in Table 2. Besides, Imwitor® 900 K has a lower melting point (≈61ºC) than Compritol® 888 ATO (≈70ºC) which favors the loading of volatile compounds.

4.       Table 3: Use 1 sig fig for Z-ave, 2 sig fig for PDI and ZP.

We have proceeded accordingly.

5.       Table 4: I think this can go in the Supplementary section.

We have proceeded accordingly. Table 4 is now Table S1.

6.       Line 303: The authors say the ZP was kept at 0 mV. How was that achieved? Seems very difficult, because that would cause the particles to sediment/flocculate. Please explain/

We have edited the paragraph as follows:As shown in Table 3, the ZP values were kept at 0 mV in all tested experiments and indicated that the independent variables (lipid and surfactant concentrations), as well as their interaction in the two levels tested, had no influence on the electrical charge of the α-pinene-loaded SLN (Figure 2C). Such low ZP values were attributed to the presence of the hydrophilic chains of poloxamer 188 (polyoxypropylene – polyoxyethylene co-polymer) which also contributed to improve the colloidal stability of SLN in dispersion, limiting the risk of aggregation.

7.       Figure 4. A – should that be SLN1? Please correct the title. Also, one figure illustrating particle sedimentation and one illustrating no sedimentation is enough. Others can be moved to supplementary section. Alternatively, can these be overlayed and shown in different colours?

We have proceeded accordingly. Only the optimized formulation has been kept as Figure 4. The remaining are now Figure S1.

8.       Instability index – do you have a formula? If so please include (replace lines 375-377 with equation). Provide reference too. This is a very good point and may be helpful for other researchers.

There is no formula as it is given by the software of LumiSizer. Reference has been provided: “This means that, for the same total clarification, samples with high clarification rates tend to be more unstable [47].

9.       Similarly, line 392, there is an equation. Provide equation numbers to all equations in the manuscript and if possible have them on a separate line.

Whenever applicable, we have proceeded accordingly. Equation has been reintroduced.

10.   Table 5: Errors? Or standard deviations should be included.

The software does not provide SD as the results translate the sedimentation velocity.

11.   Figure 6: Overlay all figures.

Figures are obtained from the software of LumiSizer and cannot be edited.

Submission Date

15 June 2019

Date of this review

25 Jun 2019 03:50:40

Reviewer 3 Report

The article titled “Development and Optimization of Alpha-pinene-loaded Solid Lipid Nanoparticles (SLN) using Experimental Factorial Design and Dispersion Analysis” describes the optimization process of alpha-pinene containing different lipid formulations through measurement of particle size, polydispersity and zeta potential. Although the article is written and explained well, the fundamental rationale behind the choice of such lipids and explanations of different observations are significantly missing. The article represents a very well documented log of experimental findings and very little can be learnt from such findings. The article also does not explain well the basis of carrying out such project, and its findings cannot be somehow correlated to any actual problem that the authors are trying to address. The introduction part does not take care of this.

Thus, this reviewer suggests a through modification and reorganization of ideas projected in this article. Also, for each experimental result authors are required to highlight the motivation and then explain the obtained result with validation or cross-validation of proposed hypotheses. It will not be sufficient to only repeat the experimental findings in words without proper scientific explanation. So, at this point, the article may not quality for publication in ‘molecules’.

Author Response

Review Report Form

Open Review

English language and style

( ) Extensive editing of English language and style required 
( ) Moderate English changes required 
(x) English language and style are fine/minor spell check required 
( ) I don't feel qualified to judge about the English language and style 

Yes

Can be improved

Must be improved

Not applicable

Does the   introduction provide sufficient background and include all relevant   references?

( )

( )

(x)

( )

Is the research   design appropriate?

( )

( )

(x)

( )

Are the methods   adequately described?

( )

( )

(x)

( )

Are the results   clearly presented?

( )

( )

(x)

( )

Are the   conclusions supported by the results?

( )

( )

(x)

( )

Comments and Suggestions for Authors

The article titled “Development and Optimization of Alpha-pinene-loaded Solid Lipid Nanoparticles (SLN) using Experimental Factorial Design and Dispersion Analysis” describes the optimization process of alpha-pinene containing different lipid formulations through measurement of particle size, polydispersity and zeta potential. Although the article is written and explained well, the fundamental rationale behind the choice of such lipids and explanations of different observations are significantly missing. The article represents a very well documented log of experimental findings and very little can be learnt from such findings. The article also does not explain well the basis of carrying out such project, and its findings cannot be somehow correlated to any actual problem that the authors are trying to address. The introduction part does not take care of this.

Thus, this reviewer suggests a through modification and reorganization of ideas projected in this article. Also, for each experimental result authors are required to highlight the motivation and then explain the obtained result with validation or cross-validation of proposed hypotheses. It will not be sufficient to only repeat the experimental findings in words without proper scientific explanation. So, at this point, the article may not quality for publication in ‘molecules’.

We have edited section introduction to address the queries of the referee, including the motivation to structure the paper as it is. Introduction is now given as follows:

Innovative, non-toxic lipid nanoparticles, such as Solid Lipid Nanoparticles (SLN), are recognised as suitable delivery carriers for lipophilic active pharmaceutical ingredients (APIs) [1, 2]. SLN are obtained from physiological and biodegradable lipids, classified as Generally Recognized As Safe (GRAS) [3, 4], being mainly composed of solid lipids (e.g. triglycerides, saturated fatty acids or waxes) [5]. The prerequisite for the selection of the raw materials is their melting point (above 40ºC), because SLN have to be solid at both of the room and body temperatures [6]. The interest in SLN for loading essential oils, containing monoterpenes, relies on their capacity to modify the release profile of perfumes and fragrances [7, 8], as well as their high tolerance for a topical application on the skin [1, 9, 10].

Monoterpenes (C10H16) are recognized of industrial interest, mainly in the field of pharmaceutics, nutraceuticals and cosmetics [2, 11, 12-20]. One of the most commonly well-known bicyclic monoterpenes [21] is alpha-pinene (Figure 1), also called 2,6,6-trimethylbicyclo [3.1.1] hept-2-ene [22, 23]. Alpha-pinene occurs in the essential oil of several coniferous trees from Pinaceae (genus Pinus) [24] and Lamiaceae family (e.g. lavender, genus Lavandula) [25, 26, 27], rosemary (genus Rosmarinus, species Rosmarinus officinalis L. [21, 25, 28]. It can also be extracted from mandarin peel oil (Rutaceae family, Citrus reticulate species) [29]. Alpha-pinene is a colorless liquid at room temperature, substantially insoluble in water, being therefore a suitable candidate for the loading in lipid nanoparticles as SLN for modified release. This monoterpene can be widely used as a raw material for the synthesis of products with a high commercial value [22, 30, 31] in the pharmaceutical, fragrance and flavor industries [22, 23]. Alpha-pinene exhibits several biological and medical properties, e.g. antimicrobial [32, 33], insecticidal or antioxidant activities [23, 34]. It has anti-inflammatory [35, 36], anti-stress and anti-convulsive activities, as well as sedative effects and antitumor activity [21, 37-41]. Several products can be obtained by submitting α-pinene to different catalytic chemical processes. For instance, α-pinene oxidation produces α-pinene oxide, verbenone and verbenol, which are used in the production of artificial flavors, fragrances and medicines [42]. Other terpenes used in industry, such as β-pinene [43], tricyclene, camphene, limonene, p-cymene, terpinenes or terpinolenes are the result of α-pinene isomerization in the presence of acid catalysts [24, 44]. Due to the identified beneficial effects of α-pinene, its loading in SLN maybe an interesting non-toxic skin formulation.

In this study, experimental factorial design was used to develop and optimize α-pinene-loaded SLN dispersion with suitable physicochemical parameters for the encapsulation of α-pinene. Additionally, the stability of α-pinene-loaded SLN stored at room temperature was characterized by using a dispersion analyzer (LUMiSizer®) with STEP-Technology® (Space- and Time-resolved Extinction Profiles).”.

Submission Date

15 June 2019

Date of this review

02 Jul 2019 04:09:06

Round 2

Reviewer 1 Report

Accept article in present form.

Reviewer 2 Report

Thanks for clarifying the points that I had raised earlier. I am satisfied with the corrections made. Although I am not sure about the novelty of the work here, the use of LumiSizer is interesting and may influence other researchers to use similar techniques and it is only for this reason I support acceptance of this manuscript. Congratulations!

Reviewer 3 Report

The authors have partly addressed the concerns. 

Visibility of all figures should be improved (axis labels, numbers and even plots- see fig. 2, 3 4, etc.) 

This manuscript is a resubmission of an earlier submission. The following is a list of the peer review reports and author responses from that submission.